# Deep learning enabled smart mats as a scalable floor monitoring system

Qiongfeng Shi [1,2,3,4], Zixuan Zhang[1,3,4], Tianyiyi He[1,3,4], Zhongda Sun [1,2,3,4], Bingjie Wang[1,3], Yuqin Feng[1,3], Xuechuan Shan [2,5], Budiman Salam[2,5] & Chengkuo Lee [1,2,3,4,6✉]

Toward smart building and smart home, floor as one of our most frequently interactive interfaces can be implemented with embedded sensors to extract abundant sensory information without the video-taken concerns. Yet the previously developed floor sensors are normally of small scale, high implementation cost, large power consumption, and complicated device configuration. Here we show a smart floor monitoring system through the integration of self-powered triboelectric floor mats and deep learning-based data analytics. The floor mats are fabricated with unique "identity" electrode patterns using a low-cost and highly scalable screen printing technique, enabling a parallel connection to reduce the system complexity and the deep-learning computational cost. The stepping position, activity status, and identity information can be determined according to the instant sensory data analytics. This developed smart floor technology can establish the foundation using floor as the functional interface for diverse applications in smart building/home, e.g., intelligent automation, healthcare, and security.

[1] Department of Electrical and Computer Engineering, National University of Singapore, 4 Engineering Drive 3, Singapore 117576, Singapore. [2] Singapore Institute of Manufacturing Technology and National University of Singapore (SIMTech-NUS) Joint Lab on Large-area Flexible Hybrid Electronics, National University of Singapore, 4 Engineering Drive 3, Singapore 117576, Singapore. [3] Center for Intelligent Sensors and MEMS (CISM), National University of Singapore, 5 Engineering Drive 1, Singapore 117608, Singapore. [4] National University of Singapore Suzhou Research Institute (NUSRI), Suzhou Industrial Park, Suzhou 215123, China. [5] Printed Intelligent Device Group, Singapore Institute of Manufacturing Technology, Agency for Science, Technology and Research (A*STAR), Singapore 637662, Singapore. [6] NUS Graduate School for Integrative Science and Engineering (NGS), National University of Singapore, Singapore 117456, Singapore. ✉email: elelc@nus.edu.sg

Under the scope of ultrafast data transmission rate promised by the new communication technologies in the era of internet of things (IoT) and fifth-generation wireless networks, numerous electronic devices with wireless interconnections with each other as well as the network cloud can be deployed within a building, enabling the realization of intelligent monitoring and response systems in the smart building/home applications[1–6]. In general, camera-based surveillance for monitoring and recognition are commonly adopted in the office and home areas, but it raises severe privacy concerns in the present society. To better protect people from the video-taken privacy issue, optical approaches such as laser beam scanning have been proposed as a potential solution[7,8]. Yet the acquired sensory information is rather limited and the laser beam is easy to be blocked by other objects, resulting in information loss and inaccurate sensing. Furthermore, the implementation and operation of such a system are highly costly and power consuming, incompatible with the sustainable development of smart building/ home. Floor, on the other hand, as one of our most frequently interactive interfaces, can be implemented with embedded sensors to acquire the abundant sensory information from human walking, including indoor position, activity status, individual identity, etc. The detected sensory information is of great importance in the aspect of elderly people nursing (e.g., fall detection by monitoring the irregular output signals in the time domain—abnormal outputs in a short period followed by no outputs), home automation of air conditioning/lighting, and security monitoring.

In terms of floor sensors, the commonly adopted transducing mechanisms include resistive, capacitive, piezoelectric, and triboelectric mechanism[9–15]. With the self-generated electrical signals in response to the mechanical stimuli, the piezoelectric and triboelectric mechanisms exhibit extra advantages such as the reduction of system-level power consumption and the potential realization of self-sustainability. However, most of the previously reported floor sensors are only demonstrated on a small scale and show low scalability in the large-area floor sensing. To cover a large area, the number of sensing pixels and signal collecting electrodes/channels needs to be dramatically increased, introducing extreme complications in the electrode layout, interconnection, and signal readout/process/analysis. Besides, large-area manufacturing and deployment cost of conventional resistive, capacitive, and piezoelectric sensors is also another major concern in the practical implementation. Hence, a low-cost and large-scale floor sensing technology with optimized design to reduce systemic complexity is highly desired to enable diverse smart building applications.

Combining the low-cost triboelectric sensing mechanism with the large-scale printing technique offers a promising solution. On one hand, triboelectric sensors can produce self-generated electrical signals based on the coupling effect of contact electrification and electrostatic induction[16–21], showing superior merits of simple configuration, great manufacturing compatibility, high scalability, no material limitation, and low cost[22–27]. On the other hand, printing techniques such as roll-to-roll printing, inkjet printing, and screen printing have been extensively adopted in large-scale device fabrication[28–31]. Thus the combination of the triboelectric mechanism and the printing technique provides a good potential to achieve low-cost, large-scale, and self-powered floor sensing technology. Subsequently, another issue to be addressed is to minimize the systemic complexity and the number of signal collecting electrodes/channels. One possible approach is arranging four electrodes at the edges of a sensing area and taking the output ratios of opposite electrodes to determine the contact position with induced triboelectric charges[32–34]. However, with a large sensing area, the induced outputs will be extremely small

due to the large coupling distance and thus not applicable in the floor sensing situation. Meanwhile, another possible approach is connecting different electrodes with distinct patterns in parallel to reduce the total electrode number and still maintain good sensing performance[35–38], based on the unique fingerprint-like signal from each electrode pattern. This sensing methodology can be considered as a potential solution to minimize the systemic complexity, yet no large-area application such as floor sensing has been demonstrated.

Currently, the functionality of most sensors is based on the time-domain data analytics of the acquired sensing signals, normally by the signal magnitude and frequency. But this preliminary analytics approach may lose some important features in the sensing signals, such as the identity information. To extract the full sensory information from sensors, advanced artificial intelligence (AI) technology using machine learning (ML)-assisted data analytics can be applied in a monitoring system. The recent technology fusion of AI and IoT has promoted the rapid development of artificial intelligence of things (AIoT) systems that can acquire, analyze, and respond to the external stimuli more intelligently, with the applied ML analytics on the sensory dataset to realize personalized authentication and object/intention identification[39–43]. It can thus be expected that, with the introduction of AI in a floor sensing system, a higher level of intelligence-enabled position monitoring, home automation, personalized healthcare, and authentication can be achieved toward the actual "smart" building/home.

Herein, deep learning-enabled smart mats (DLES-mats, i.e., floor mats) based on the triboelectric mechanism are developed to realize an intelligent, low-cost, and highly scalable floor monitoring system. The smart floor monitoring system is achieved through the integration of a minimal-electrode-output triboelectric floor mat array with advanced deep learning (DL)-based data analytics. The DLES-mats are fabricated by screen printing, exhibiting the merits of cost-effectiveness, high scalability, and self-sustainability in large-area applications. A distinct electrode pattern with varying coverage rate is designed for each DLES-mat, mimicking the unique identification of the QR (quick response) code system. Thus, after the parallel connection in an interval scheme, minimal two-electrode outputs with distinguishable and stable characteristics for the whole DLES-mat array can be achieved. The differentiation of the parallel-connected DLES-mats is based on the relative magnitude of output signals, enabling indoor positioning and activity monitoring. Furthermore, with the integrated DL-based data analytics, identity information associated with walking gait patterns can be extracted from the output signals using the convolutional neural network (CNN) model. Meanwhile, benefited from the minimal two-electrode outputs, huge computing resources can be saved compared to the traditional image or massive channel-based process, enabling faster data analytics for real-time applications in smart building/home.

## Results

**Minimal-electrode design and operation mechanism.** A potential application scenario of the smart floor monitoring system is shown in Fig. 1a, where the DLES-mat array is attached onto the corridor floor. When a person is walking on the DLES-mat array, the generated electrical signals from the contact–separation motion of each stepping can be acquired and then used for position sensing of the person. Accordingly, the corridor light above the corresponding position can be switched on by the system for lighting purpose. When no signal is detected for a certain period of time, the system will then switch off the lights for energy-saving purpose. With the integrated DL-based

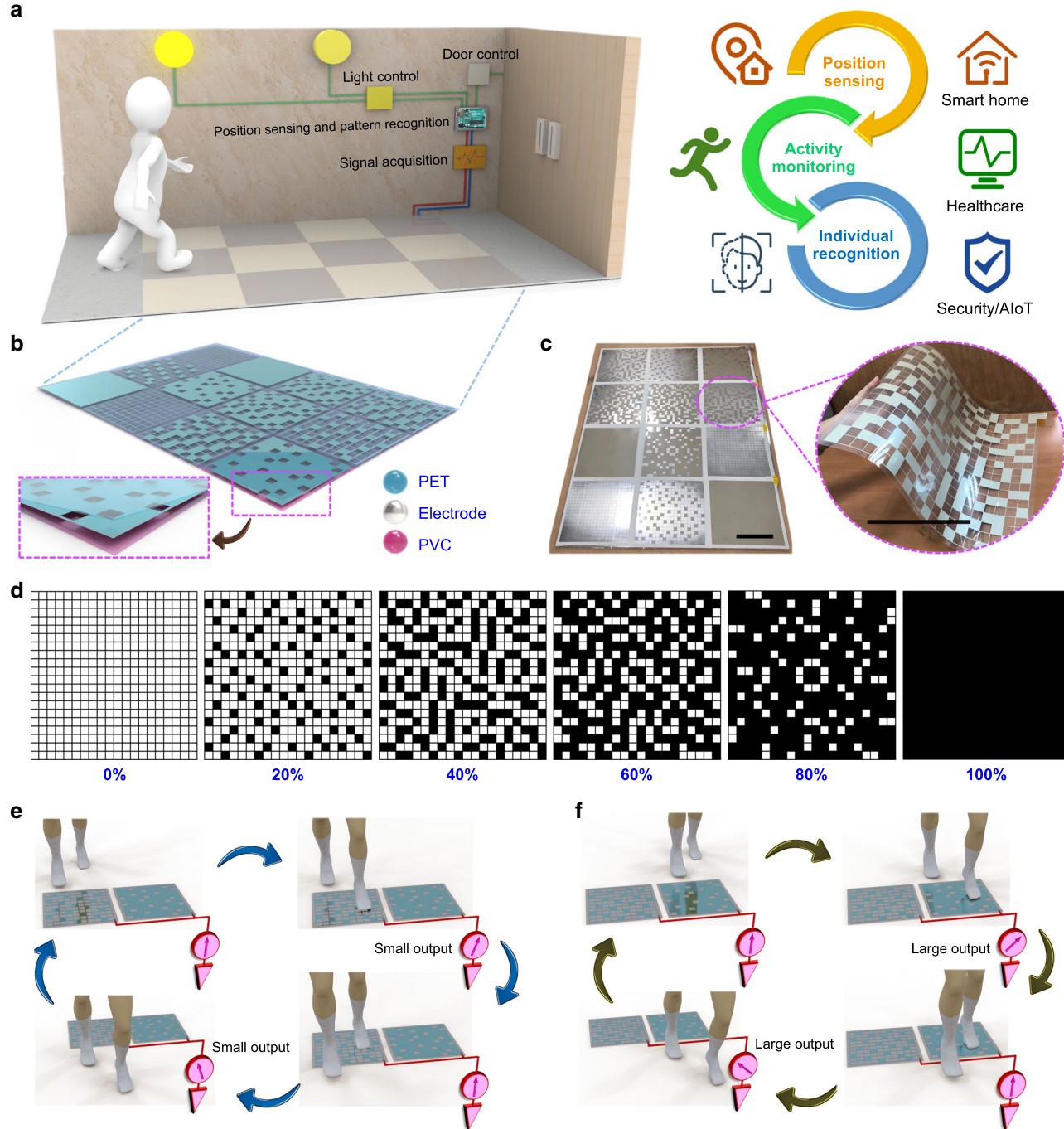

**Fig. 1 The smart floor monitoring system based on the deep learning-enabled smart mats (DLES-mats). a** The conceptual diagram of the smart floor monitoring system and its potential applications of position sensing, activity monitoring, and individual recognition in the smart building/home scenarios. **b** The assembled triboelectric DLES-mat array with a 3 × 4 arrangement, where the inset shows the three device layers fabricated by a low-cost and large-scale screen printing technique. **c** The digital photographs of the floor mat array and an individual floor mat with a 40% electrode coverage rate, where the scale bars are both 20 cm. **d** The detailed electrode layout of the floor mats with different electrode coverage rates from 0 to 100% to achieve distinguishable signal patterns after parallel connection. **e**, **f** The operation mechanism of the parallel-connected floor mat array when a person walks on/off a floor mat with **e** smaller electrode coverage rate and **f** larger electrode coverage rate, where the output signals with relatively smaller and higher magnitude are generated, respectively.

data analytics, individual recognition can also be achieved according to different walking gaits. The realized individual recognition can be adopted in the automatic access by opening the door for recognized valid users. Therefore, the smart floor monitoring system with the superior capability of position sensing, activity monitoring, and individual recognition exhibits great potential toward realizing the smart building/home

applications in the aspects of automation, healthcare, security, and AIoT.

Through the parallel connection of multiple sensors, the number of required electrodes in the system can be effectively reduced. But to differentiate the output signals, each sensor should possess a distinct characteristic, like the identity of a QR code. According to the triboelectric theory, under the same

contact conditions (e.g., contact area, pressure, etc.), the same amount of triboelectric charges should be generated on a dielectric friction surface, and a larger electrode area beneath the dielectric surface can collect more charges through the electrostatic induction. Normally, the triboelectric sensor can be analyzed by a variable capacitor model, with its generated open-circuit voltage given by $V_{OC} = Q/C$, where $Q$ is the effective induced charges on the electrode (positively related to the electrode area) and $C$ is the equivalent capacitance of the triboelectric sensor. For parallel-connected triboelectric sensors, they share the same equivalent capacitance in the output generation. In this regard, triboelectric sensors with different electrode areas will generate outputs of different magnitudes, proportional to the effective induced charges on the electrode, making them distinguishable in a parallel connection. Therefore, in the smart floor monitoring system, DLES-mats with different electrode coverage rates are designed and fabricated through screen printing the designated electrode patterns on a poly-ethylene terephthalate (PET) film and further packaging with another polyvinyl chloride (PVC) film. The schematic diagram of the as-designed DLES-mat array is shown in Fig. 1b, where the enlarged image depicts the three stacking device layers, i.e., PET friction layer, silver (Ag) electrode layer, and PVC substrate layer. Figure 1c illustrates the photographs of the assembled floor mat array as well as the flexibility of a floor mat (40% electrode coverage rate). The detailed electrode patterns of the floor mats are presented in Fig. 1d, where six electrode coverage rates (from 0 to 100%) with 20% difference are designed to achieve a rational balance between the clear distinction and the number of floor mats. The black color represents the printed Ag electrode, with the uniform grid lines as the electrode interconnection through-out the floor mat and the filled squares to obtain different coverage rates. The vertical and horizontal grid electrode lines form a 20 × 20 array of empty squares (20 mm × 20 mm), that can be further selected to be filled with Ag to achieve different and uniformly distributed electrode coverage rates. It is worth to mention that all the electrode patterns of 20, 40, 60, and 80% can be obtained from the same printing mask of 20%. First, with the mask oriented upward, the printing results in the 20% DLES-mat. Afterward, a further printing on the 20% DLES-mat with the mask oriented downward leads to the 40% DLES-mat. Similarly, subsequent printing with the mask oriented right and left will result in the 60% and the 80% DLES-mats, respectively. The gradual printing results of these four DLES-mats with different mask orientations are shown in Supplementary Fig. 1. Thus, with this design, only one printing mask is required for these four DLES-mats, contributing to the cost reduction in the DLES-mat fabrication. The cost estimation of the fabricated DLES-mats can be found in Supplementary Note 1.

To elucidate the operation mechanism of the assembled DLES-mat array in a straightforward manner, the configuration with two DLES-mats in parallel connection is used as an example. Since the PET friction layer adopted here is relatively positive, most common materials (e.g., socks and shoe soles) become negatively charged after contacting with it, leaving its surface positively charged. After that, when a person steps on the DLES-mat with less electrode coverage rate as shown in Fig. 1e, a certain amount of electrons will be repelled to flow through the external circuit to the ground until new electrostatic equilibrium is achieved. According to the above theoretical model, the amount of flowing electrons is proportional to the electrode coverage area. Thus stepping on the DLES-mat with less electrode coverage rate generates a smaller output current/voltage pulse. When the person steps off the DLES-mat, the same amount of electrons flow back to the electrode from the ground, generating a reverse current/voltage pulse in the external circuit. On the other hand,

when the person steps on and off the DLES-mat with higher electrode coverage rate (Fig. 1f), a larger amount of electrons will flow in the external circuit, thus producing a larger output current/voltage pulse. A more comprehensive illustration of the operation mechanism with detailed charge transfer processes corresponding to different walking stages is shown in Supplementary Fig. 2. Through the design of varying electrode coverage rates, the generated triboelectric signals with different relative magnitudes can thus be adopted to distinguish the outputs from different DLES-mats and determine the corresponding walking positions as well.

**Characteristics of DLES-mat output and connection scheme.** With the parallel connection of the six fabricated DLES-mats (0–100%), the output from each floor mat is first characterized with repeated stepping motions by both the right foot and the left foot wearing shoes with polytetrafluoroethylene (PTFE) sole in four directions (i.e., N, north; E, east; S, south; W, west). The generated output voltages on a 1 MΩ external load from the six DLES-mats are shown in Fig. 2a–f, respectively. Figure 2g summarizes the average output trends of the maximum voltage, minimum voltage, and peak-to-peak voltage. It can be clearly observed that, as expected, the absolute magnitudes of all these three output voltages show a positive relationship with the electrode coverage rate. To maximize the distinction between the outputs of different DLES-mats, the peak-to-peak voltages are used for signal analysis and later characterization. Next, the effect of different users wearing PTFE shoes on the output performance is investigated, as depicted in Fig. 2h. Although the output magnitudes of different users are different, the increment trend of relative output magnitudes of each user is similar, suggesting the suitability of the DLES-mat design in individual position sensing. The position of a user on the DLES-mats can be determined by the relative output magnitudes with respect to that from the 0% DLES-mat. In addition, the effect of different contact materials is also investigated through the same user wearing different materials, i.e., cotton sock, ethylene vinyl acetate shoe, and PTFE shoe. As indicated in Fig. 2i, similar increment trends of relative output magnitudes can be observed from all the materials, once again showing the applicability of the floor mat design in the scenario of position monitoring. When the generated output voltages are measured on a 100 MΩ load, similar results can be achieved, as presented in Supplementary Fig. 3. It is worth noting that higher absolute magnitude is achieved with the 100 MΩ load due to its higher resistance, but the relative output magnitudes exhibit similar trends. On the other hand, because of the higher measuring resistance, the generated output pulses on the 100 MΩ load take longer time to discharge (RC discharge), causing the overall signals to shift upward in each repeating period of stepping. The detailed output waveforms on the 1 and 100 MΩ load with different discharge time can be found in Supplementary Fig. 4.

With the unique electrode pattern associated with each DLES-mat, they can be connected in parallel to effectively reduce the number of output electrodes. Although clear differentiation can be achieved when stepping on the individual DLES-mat, the performance of the DLES-mat array under subsequent walking motions still needs further investigation. Thus a parallel connection of 12 DLES-mats (2 sets of 0–100%) in one-dimensional arrangement is constructed, as depicted in Fig. 3a. When a person subsequently walks on the 12 DLES-mats, the generated voltages on a 1 MΩ load and a 100 MΩ load are illustrated in Fig. 3b, d, respectively. PTFE is adopted as the contact material here and hereafter unless otherwise specified. Two rounds of forward–backward walking are repeated with left foot stepping first for the first round and right foot stepping first

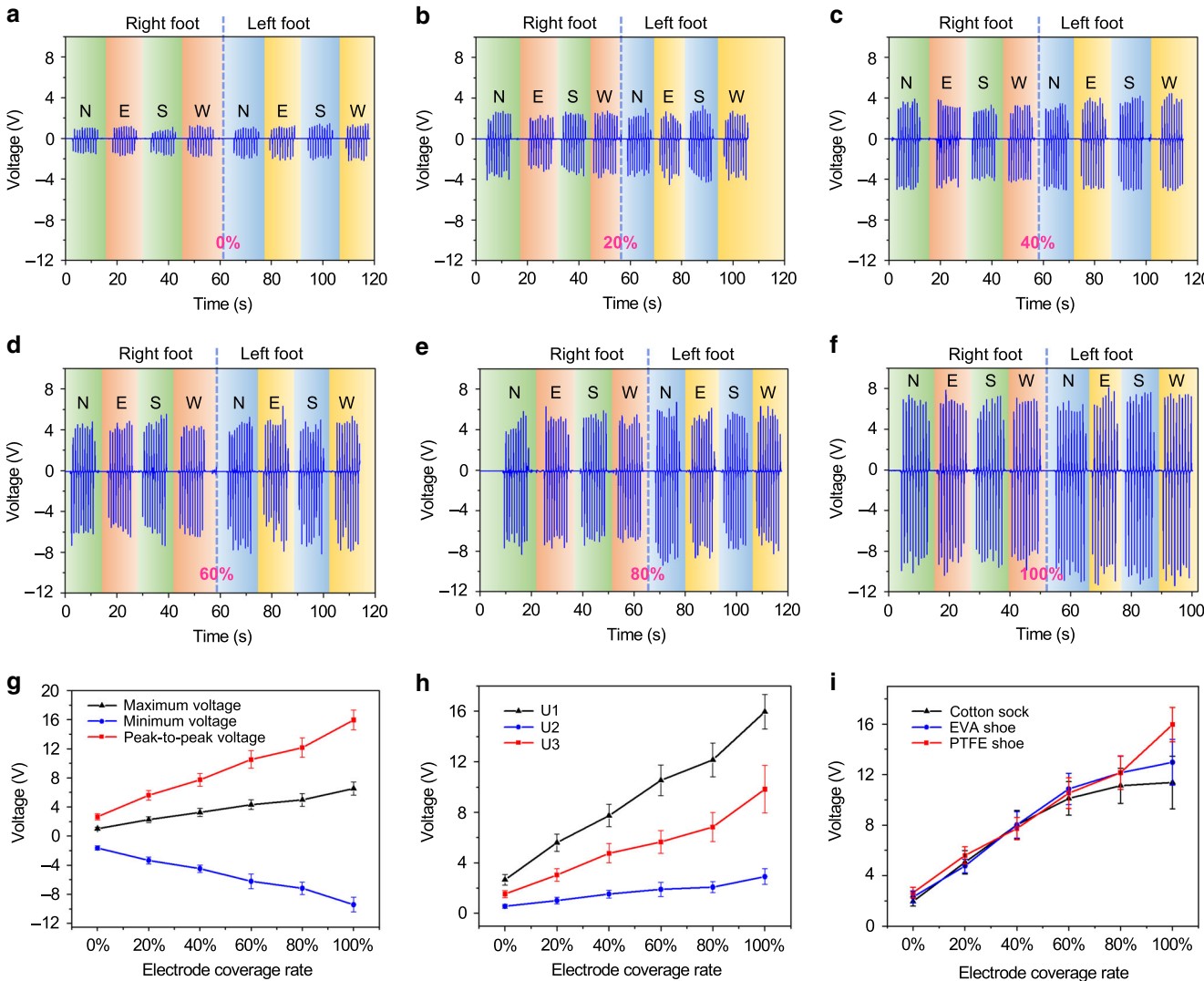

**Fig. 2 Characteristics of the output performance of the individual DLES-mat on a 1 MΩ external load. a–f** The generated voltages by repeated stepping motions with polytetrafluoroethylene (PTFE) shoes (four directions toward the north, east, south, and west for both the right foot and the left foot) on DLES-mats with an electrode coverage rate of **a** 0%, **b** 20%, **c** 40%, **d** 60%, **e** 80%, and **f** 100%. **g** The corresponding maximum voltage, minimum voltage, and peak-to-peak voltage from Fig. 2a–f with respect to different electrode coverage rates, showing clear increment trends with the increased electrode coverage rate. The error bars indicate the standard deviation. **h** The effect of different users wearing PTFE shoes on the output peak-to-peak voltage. Similar increment trends of relative magnitudes can be observed from different users, indicating the suitability of the DLES-mat design for individual position sensing. **i** The effect of different contact materials worn by the same user on the output peak-to-peak voltage, where similar increment trends can also be observed.

for the second round. It can be seen that the output voltage on the 100 MΩ load has a wider pulse width due to its much slower discharge time, leading to the overlapping of adjacent voltage pulses and the distortions of output signals. The corresponding peak-to-peak voltages extracted from Fig. 3b, d are plotted in Fig. 3c, e, respectively. A more stable increment–decrement trend of relative magnitudes can be observed for the 1 MΩ load due to its rapid discharge time compared to the 100 MΩ load. Yet the resultant voltage trend is still unsatisfactory, with a clear deviation from the ideal increment–decrement trend. This deviation is caused by the overlapping of two opposite voltage pulses from two simultaneous stepping motions, i.e., a negative pulse from stepping on the next DLES-mat and a positive pulse from stepping off (leaving) the previous DLES-mat. Therefore, to improve the signal stability of the detected output voltages, an interval parallel connection is implemented and investigated (Fig. 3f). From the measurement results shown in Fig. 3g–j, the

corresponding output voltages exhibit a much more stable increment–decrement trend for both the 1 MΩ load and the 100 MΩ load, since the interference from the walking motions on adjacent DLES-mats is eliminated. That is to say, one walking motion (stepping on and stepping off) can be fully completed on one DLES-mat before entering the next DLES-mat connected with the same output electrode. Thus no overlapping of voltage pulses is introduced in the generated signal waveforms and a more ideal increment–decrement trend can be realized. Similarly, the connection scheme for one set of DLES-mats with the parallel and interval parallel connection, is also investigated in Supplementary Fig. 5. The same conclusion can be drawn that the interval parallel connection produces a more stable and ideal increment–decrement trend for signal detection and analysis. Hence, the interval parallel connection scheme with two output electrodes is adopted in the DLES-mat array configuration to effectively reduce the number of required sensing electrodes and

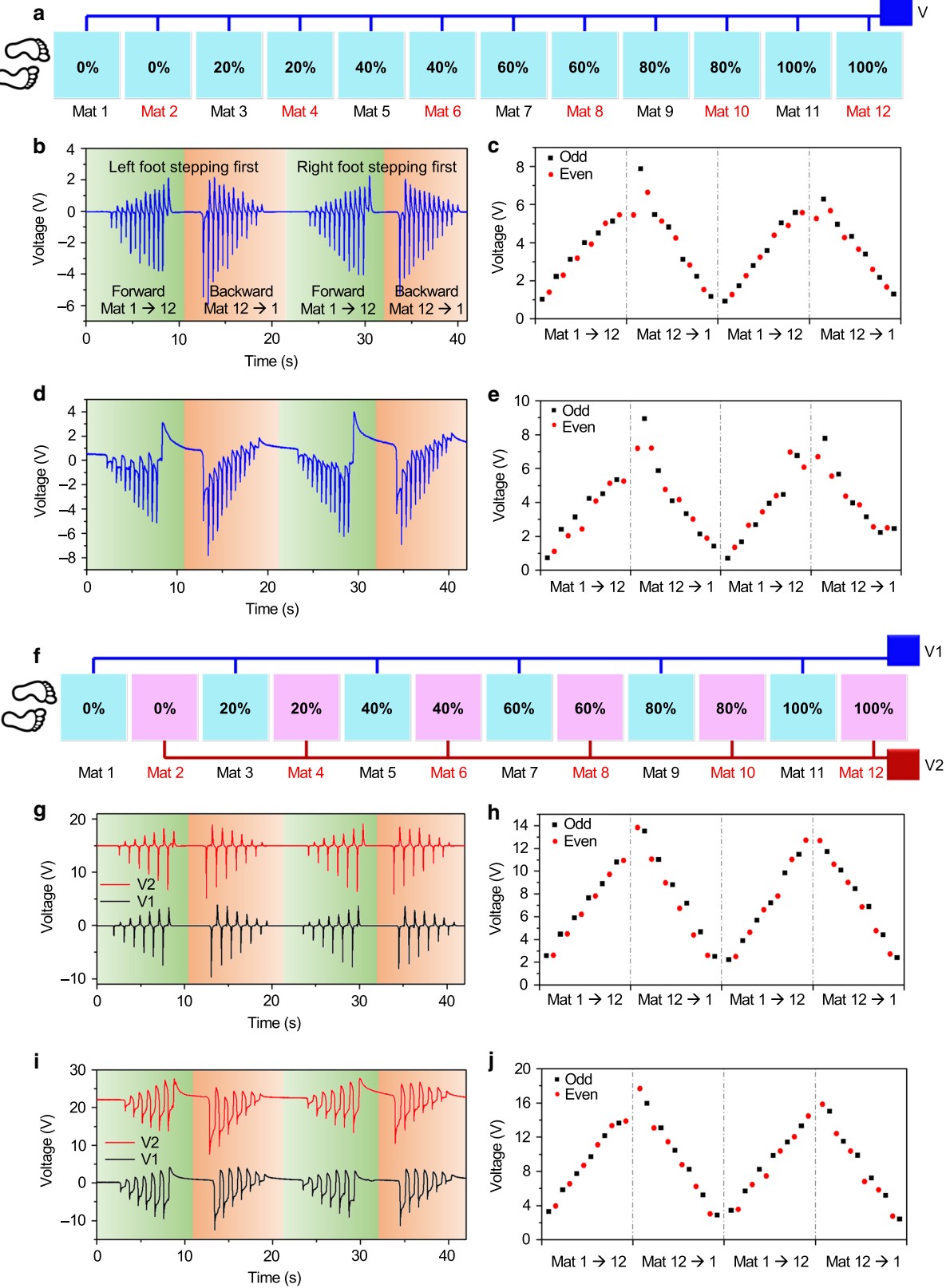

**Fig. 3 Investigation of the electrode connection scheme. a** The schematic diagram of the parallel connection of 12 DLES-mats into one output electrode. **b**–**e** The generated output voltages and corresponding peak-to-peak voltage magnitudes on **b**, **c** a 1 MΩ and **d**, **e** a 100 MΩ external load with two rounds of forward–backward walking. Large signal distortions and magnitude deviations from the ideal situations are found due to the overlapping of two opposite voltage pulses caused by simultaneous walking on the next DLES-mat and walking off the previous DLES-mat. **f** The schematic diagram of the interval parallel connection of all the DLES-mats into two output electrodes to address the above issue. **g**–**j** The generated output voltages and the corresponding peak-to-peak voltage magnitudes on **g**, **h** a 1 MΩ and **i**, **j** a 100 MΩ external load with two rounds of forward–backward walking. More ideal signal patterns and relative magnitudes can be achieved with the interval parallel connection scheme.

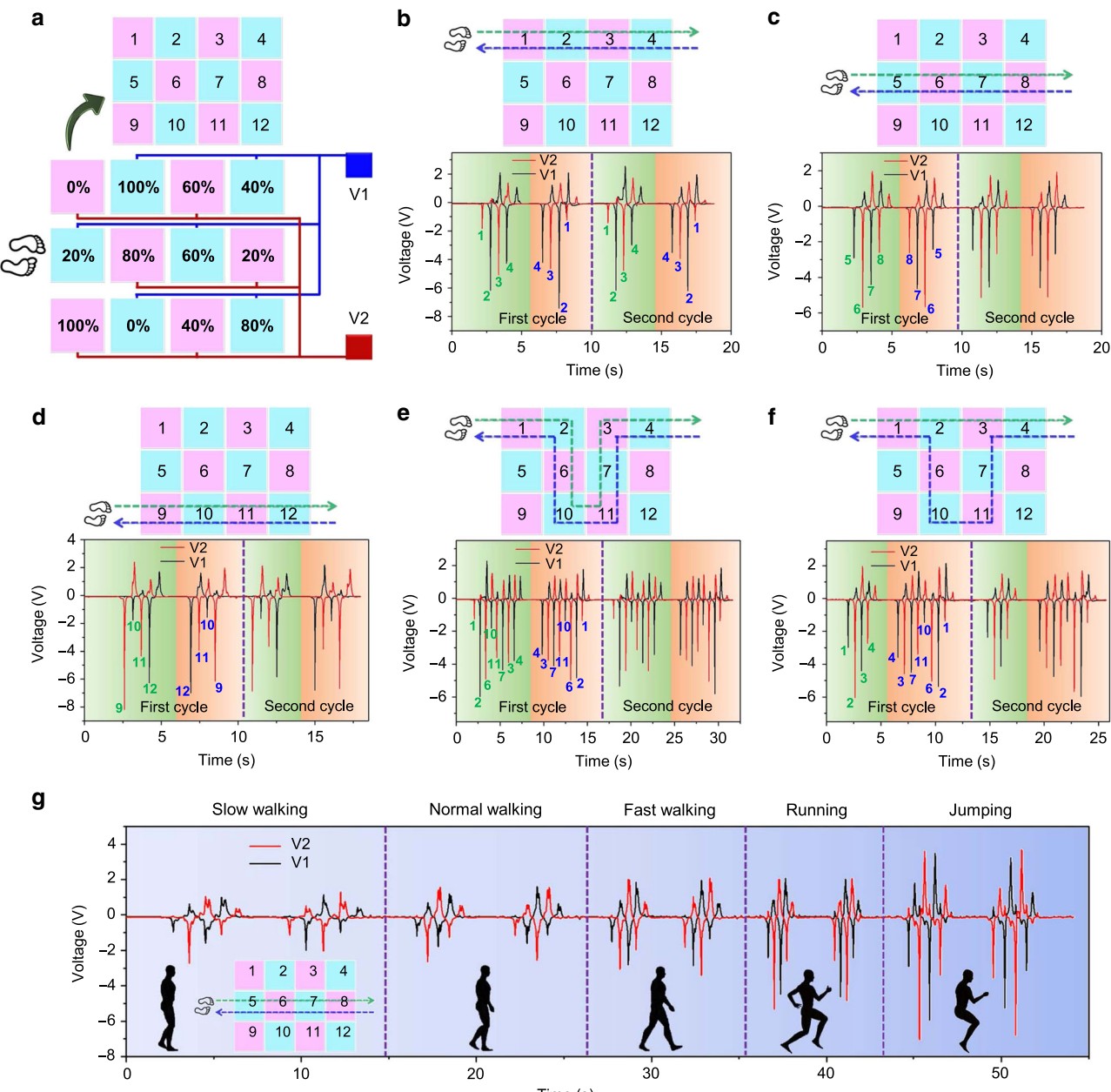

**Fig. 4 DLES-mat array for position/trajectory detection and activity monitoring. a** The schematic diagram of the constructed 3 × 4 DLES-mat array with two output electrodes. The 12 digits (1–12) indicate the numbering for each DLES-mat in the corresponding position. **b–f** The corresponding generated voltages by the indicated two cycles of forward–backward walking trajectories on the DLES-mat array. Based on the relative signal magnitudes on the two electrodes, walking patterns on the DLES-mat array can be determined for walking trajectory detection. **g** Activity monitoring for slow walking, normal walking, fast walking, running, and jumping, according to the output signal magnitudes and frequencies.

simultaneously achieve the desired differentiation between different DLES-mats.

**DLES-mat array for position sensing and activity monitoring**. Based on the above characterizations, the interval parallel connection scheme is adopted to implement the two-dimensional DLES-mat array for actual position sensing application, as illustrated in Fig. 4a. Each set of the six floor mats is connected to one electrode, resulting in only two output electrodes required for the whole 3 × 4 array. The 12 digits (1–12) in the schematic indicate the numbering for easy references of the DLES-mats in the corresponding positions. Walking tests with two repeated forward–backward cycles (left foot stepping first in the first cycle

and right foot stepping first in the second cycle) along different trajectories are conducted to verify the position sensing capability of the DLES-mat array, as shown in Fig. 4b–f. According to the relative magnitude trend obtained in the previous study, voltage pulse with higher peak-to-peak magnitude is generated from the DLES-mat with a higher electrode coverage rate. For example, in Fig. 4b, the person is walking on the first row of the DLES-mat array, with Mat 1 (0%) and Mat 3 (60%) connected to electrode 2 (E2) and Mat 2 (100%) and Mat 4 (40%) connected to electrode 1 (E1). From the output voltages in both cycles, a clear trend can be observed in the voltage magnitudes on both electrodes. For walking forward from Mat 1 to 4, an output voltage pulse with the lowest magnitude is first generated on E2 (0%), and then an

output voltage pulse with the highest magnitude is generated on E1 (100%). Next, an output voltage pulse with a relatively smaller magnitude is generated on E2 (60%). Last, an output voltage pulse with an even smaller magnitude but still larger than the first pulse is generated on E1 (40%), until the person walks out of the DLES-mat array and no output voltage is generated. A reverse sequence of output voltages can be observed for walking backward from Mat 4 to Mat 1. The same output voltage trends can be observed in both cycles, indicating the stability of the DLES-mat array for position sensing. Similarly, for the other walking trajectories shown in Fig. 4c–f, position sensing and walking trajectory detection can also be achieved based on the generated voltage signals on both the output electrodes and their relative magnitudes. This real-time position sensing capability enables the DLES-mat array in the application scenarios of automation control (such as lighting and air conditioning) and fall detection (by detecting the abnormal signal patterns of multiple peaks in a short period due to the falling-induced rapid contacts and no outputs in the following). The typical signal patterns for the normal walking process and the walking–falling process are shown in Supplementary Fig. 6.

In addition to position sensing, the DLES-mat array can also be adopted for activity monitoring and potential energy harvesting from our daily activities. Figure 4g depicts the output signals of a person performing five different types of activities, i.e., slow walking, normal walking, fast walking, running, and jumping, on the middle row of the DLES-mat array in a forward–backward manner. Different types of activities can be easily distinguished based on the overall magnitude and time period (frequency) of the output signals, indicating the activity monitoring capability. In this regard, the DLES-mat array can be applied for potential healthcare applications in exercise monitoring, including the type of exercises, the time period of the exercise, and the burned calories based on the type and period of the exercise. Next, the output voltage and power of the DLES-mat array with respect to different external resistances are measured (Supplementary Fig. 7a–c) under normal walking. A maximum output power of 8.57 μW can be obtained at 1.96 MΩ. Due to the large capacitance of the DLES-mat array, the saturated output voltage (close to the open-circuit voltage) for the same stepping motion is relatively low according to $V_{OC} = Q/C$. With a smaller DLES-mat area/capacitance, the saturated voltage can be improved for more effective energy harvesting. Thus the outputs from individual DLES-mat (100%) of 40 cm × 40 cm and 30 cm × 12 cm are also measured (Supplementary Fig. 7d–i). The saturated voltage is greatly improved with a smaller area, but the matched resistance also increases. For the 40 cm × 40 cm and the 30 cm × 12 cm DLES-mat, the maximum output voltage is 55.0 and 144.0 V at 100 MΩ, respectively, while the maximum output power is 169.46 μW (at 9.10 MΩ) and 800.84 μW (at 13.79 MΩ). In practical applications, the rectified output voltages can be applied to charge up capacitors as sustainable power sources for other IoT devices in smart buildings. The capacitor charging and wireless sensor powering by the three devices are depicted in Supplementary Fig. 8. After charging up a 27 μF capacitor to 8 V, the stored energy is sufficient to support one operation cycle of the sensor. These results demonstrate that the operation of IoT devices with intermittent functionalities can be supported by the developed DLES-mats.

## DL-based data analytics and smart building demonstration.
Alongside with the rapid advancement of AI technology, an increasing number of AI integrated smart systems have also been developed to achieve intelligent decision-making and control automation. DL as a sub-field of ML can provide an efficient way

to automatically learn the representative features from collected raw signals by training an end-to-end neural network, which has made great achievements in analyzing the image, video, speech, and audio. When integrating the DLES-mat array with DL-assisted signal analytics, a smart floor monitoring system for not only position/activity sensing but also individual recognition can be realized. Since the walking gait pattern of a person is different from others, it can generate a unique output signal for individual recognition. The overall structure of the smart floor monitoring system is shown in Fig. 5a. When a person is walking through the DLES-mat array, triboelectric output signals are generated by the periodic contact–separation motions of human steps. These generated signals are then acquired by the signal acquisition module in an Arduino MEGA 2560 microcontroller. In terms of the training data for individual recognition, the signal data from each channel is recorded with 1600 data points (2 channels in total) and 100 samples are collected for each user (80% for training and 20% for testing). A whole dataset is built from 10 different users, with a total number of 1000 samples. The typical output voltages from each user walking through the middle row of the DLES-mat array are shown in Supplementary Fig. 9. In this study, the DL model is created based on CNN in order to provide high recognition performance, where the parameters used to construct the CNN model are labeled in Fig. 5b and Supplementary Table 1. After the training process in the CNN model with 50 training epochs, the maximum accuracy can be achieved, and the CNN model is able to generalize enough to avoid over-fitting as shown in Supplementary Fig. 10. The average recognition accuracy is 96.00% (Fig. 5c), providing great potential for high-accuracy control based on the DL prediction. In addition, recognition testing of the same user in different passing statuses (i.e., normal walking, fast walking, and running) is also conducted to demonstrate the applicability of the smart floor monitoring system in various situations. The corresponding signal patterns of four different users and the predicted results are shown in Supplementary Figs. 11 and 12, respectively. It can be observed that the trained DL model is able to distinguish the different passing statuses of the 4 users (12 classes) with an accuracy of 89.17%. Besides, if all the passing statuses from the same user are set as one individual label (just distinguish the user without knowing his passing status), the accuracy of the testing set after training reaches 91.47%. These results indicate that, even when the user passes through the DLES-mat array in different ways, the smart floor monitoring system can still recognize and identify the user with a high accuracy of 91.47%.

To demonstrate the practical usage scenarios, a virtual corridor environment mimicking the real corridor is built to reflect the real-time status of a person on the DLES-mat array, including position sensing through the peak detection and individual recognition through the DL prediction. Unlike the camera-based monitoring that normally involves the video-taken concerns, this smart floor monitoring system using a digital twin of the person in the virtual environment only shows the position information and the recognized identity, which are basic parameters required for automation, healthcare, and security applications. The overall flow of the signal acquisition and analysis process is shown in Fig. 5a. When a person first steps on the DLES-mat array on Position 1, a small negative peak is generated from E1 (20% mat), which can be adopted as the trigger signal to move the digital twin to the first DLES-mat and turn on the corresponding Light 1, as indicated in Fig. 5d. When the person continues walking, a large negative peak from E2 (80% mat, as the trigger signal to move the digital twin to the second DLES-mat) and a small positive peak from E1 (20% mat) are generated. Then upon stepping on Position 2, a negative peak with relatively smaller magnitude than the 80% mat is generated from E1 (60% mat),

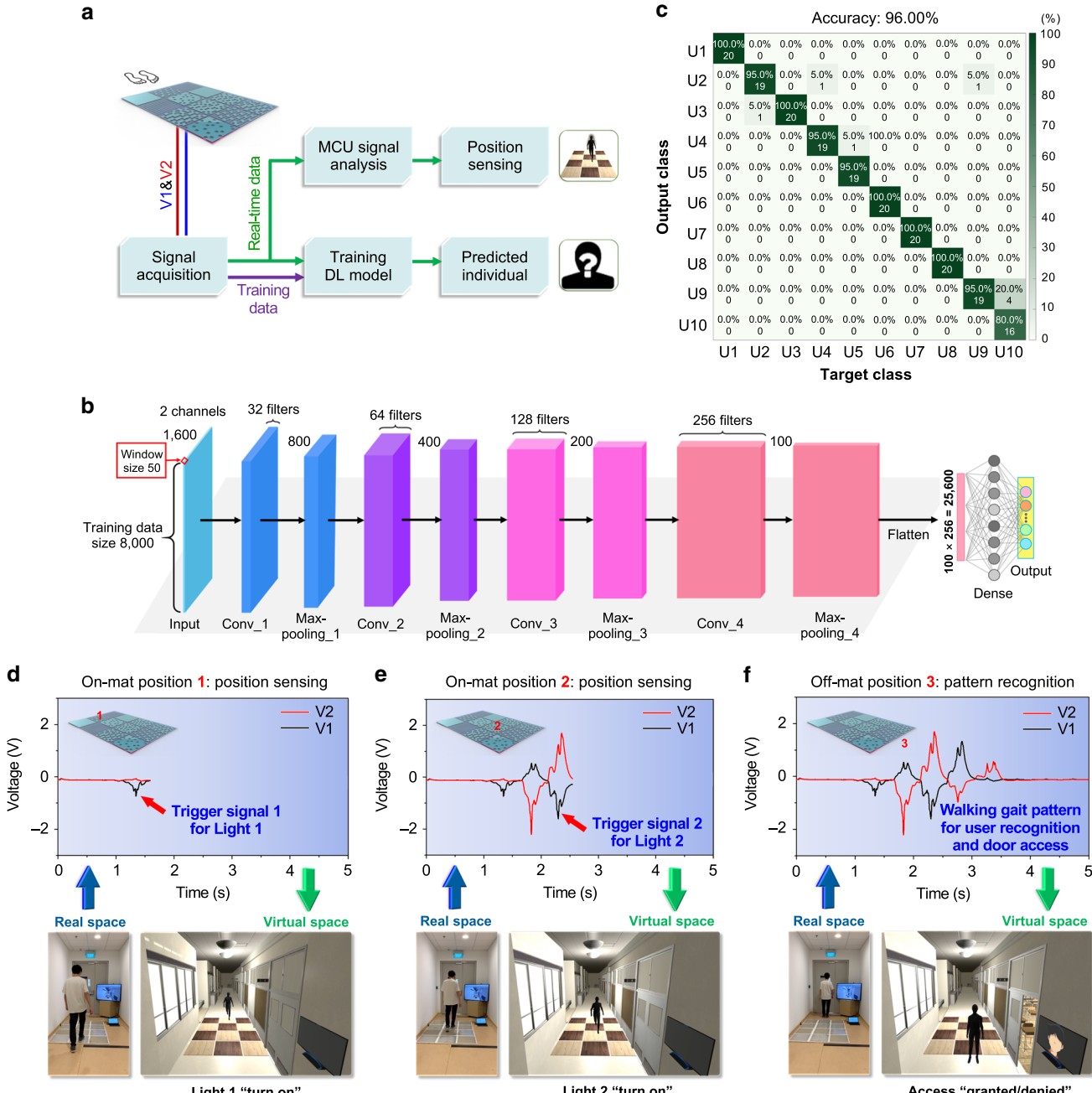

**Fig. 5 Smart floor monitoring system with integrated deep learning-assisted data analytics. a** The overall structure and data flow of the smart floor monitoring system for real-time position sensing and individual recognition in smart building/home applications. **b** The detailed structure of the convolutional neural network (CNN) training model. **c** The confusion matrix for individual recognition of 10 different users, showing a high accuracy of 96%. **d**–**f** Demonstration of the different stages in real-time position sensing and individual recognition, where a person is walking in the real space while his digital twin is controlled to walk in the virtual space correspondingly: **d** At Position 1, the first negative peak is detected and used to turn on Light 1; **e** At Position 2, the third negative peak is detected and used to turn on Light 2; **f** At Position 3, the full walking signal is detected and analyzed in CNN model for individual prediction ("valid/invalid user") and door access control ("granted/denied").

which is used as the trigger signal to move the digital twin to the third DLES-mat and turn on the corresponding Light 2, as indicated in Fig. 5e. After walking through the whole DLES-mat array and reaching Position 3, a full cycle of the output signal is generated from both electrodes as illustrated in Fig. 5f, reflecting the unique walking gait of the person. The full-cycle output signal is then analyzed by the trained DL model to predict whether the person is a valid user of the room. If the person is a valid user, then access to the room is granted and the door will be open. Otherwise, access is denied and the door will remain closed. A

video demo with three persons (two valid users and one invalid user) walking through the DLES-mat array can be seen in Supplementary Movie 1. There is a small delay between the motions in the real and virtual space due to the time taken for signal processing and analysis. As in this scenario the personal identity is still revealed with certain privacy concerns, another approach can be implanted to better protect privacy where only the recognition of valid and invalid users is required. At the training stage for the DL model, labels with privacy information like the name of the person will not be included but only a label of

"valid user" for all the users with valid access. Thus, when a person walks on the DLES-mat array with a recognized walking pattern, a message of "valid user" will be displayed without revealing any of his privacy information and the door will be automatically opened. Then if his walking pattern is not recognized, the message of "invalid user" will be displayed and the door will remain closed. In this way, the recognition of valid and invalid users can be achieved without revealing the identity and the privacy information of the person. Overall, in this demonstration, real-time position and individual recognition of a person walking on the DLES-mat array can be successfully achieved, showing the great potential of the smart floor monitoring system in smart building relative automatic control and security access.

## Discussion

In summary, a smart floor monitoring system is developed for indoor positioning, activity monitoring, and individual recognition toward the smart building/home applications. It is realized through the system integration of self-powered triboelectric DLES-mats and advanced DL-based data analytics. Benefited by the screen printing manufacturing and triboelectric sensing mechanism, the DLES-mats possess the grand advantages of low cost, high scalability, and self-sustainability that are ideally suitable for large-area floor monitoring. In addition, the design of a distinct electrode pattern enables the interval parallel connection of different DLES-mats, resulting in minimal two-electrode outputs with clear and stable differentiation for a $3 \times 4$ DLES-mat array. Furthermore, after data analytics in the developed CNN model, a smart floor monitoring system can be achieved for real-time position sensing and identity recognition. The position sensing information from each step is adopted to control the lights in corresponding positions, while the full walking signal is analyzed by the CNN model to predict whether the person is a valid user of the room so as to auto-control the door access. Comparing with camera and smart tag-based individual recognition, the smart floor monitoring system based on the dynamic gait-induced output signals provides a video-privacy-protected, highly convenient, and highly secure recognition approach. For a 10-person CNN model with 1000 data samples, the average prediction accuracy can reach up to 96.00% based on their specific walking gaits, offering a high accuracy in the practical real-time scenarios. Therefore, the developed smart floor monitoring system with the excellent capability of position sensing, activity monitoring, and identity recognition exhibits promising potential in the applications of automation, healthcare, security, and AIoT toward smart building/home.

## Methods

**Fabrication and implementation of the DLES-mats**. A thin layer of PET with relatively high triboelectric positivity is utilized as the friction surface for common foot stepping. The PET is a semi-crystalline polymer film with a high optical transparency, a thin thickness of 125 μm, and a glass transition temperature of 81.5 °C. First, to achieve individual floor mat, the large-area PET thin film is cut into a square shape with a dimension of 42 cm × 42 cm. The PET thin film is then pretreated on one side with a primer treatment for promoting the adhesion with the later printed electrode layer. After that, a layer of silver paste as the charge collection electrode is printed on the pretreated PET surface by screen printing, followed by a thermal curing at 130 °C for 30 min using a thermal oven. The printed thickness of the silver electrode is about 15 μm. Next, the PET film with the printed silver electrode is cold-laminated with a layer of 80-μm-thick PVC. The PVC layer serves as the supporting substrate with a square opening of 2 cm × 2 cm on the connector pad for wiring purpose. Following that, a copper wire is connected to the electrode by conductive paste through the opening, which is then sealed with thin Kapton tape. Last, different fabricated floor mats are pasted on a woolen floor, and the wires from each floor mat are connected based on the investigated connection scheme for later characterizations.

**Electrical characterization of the DLES-mats**. The output voltages of the triboelectric DLES-mats are measured by an oscilloscope (Agilent DSO-X3034A) with a recording impedance of 1 MΩ as well as 100 MΩ for waveform comparison. In terms of the voltage and power characteristics versus varying resistor loads, the output voltages on different loads are measured by a Keithley 6514 Electrometer connected in parallel. Then the peak power on the corresponding resistor load is calculated using the formula $P = V^2/R$, where $P$, $V$, and $R$ are the peak power, measured output voltage, and resistance of the resistor load, respectively. As for the capacitor charging, the voltages on different capacitors are also measured using the Keithley 6514 Electrometer in parallel connection with the capacitors.

**Data collection and DL training model**. The generated triboelectric signals from the DLES-mat array are acquired by the signal acquisition module in an Arduino MEGA 2560 microcontroller in a real-time manner. In terms of the training data for individual recognition, the signal data from each channel is recorded with 1600 data points (2 channels in total) and 100 samples are collected for each user's walking pattern, where 80 samples are used for training (80%) and 20 samples are used for testing (20%). A whole dataset is built from 10 different users, with a total number of 1000 samples. The CNN models used in the system are configured as follows: the categorical cross-entropy function is applied as the loss function, adaptive moment estimation (Adam) is used as the update rule due to its optimization convergence rate, and prediction accuracy is used to evaluate the model training. The CNN models are developed in Python with a Keras and TensorFlow backend. The feature-based models are trained on a standard consumer-grade computer. The learning rate can be adjusted during training using a Keras callback.

**Demonstration of the smart building application**. For the real-time demonstration, the two-channel triboelectric signals from the DLES-mat array are first connected to the analog input ports of an Arduino MEGA 2560 microcontroller. The acquired electrical signals are sent to a laptop by USB cable communication instantly. Following that, the received signals are processed in Python for peak detection and pattern recognition. When the first negative peak is detected, a control command is sent to Unity 3D through TCP/IP communication to move the digital twin to the first DLES-mat position and turn on Light 1 in the virtual scene. Subsequently, for the following detected negative peaks, the digital twin is controlled to move to the corresponding positions. Light 2 is turned on when the third negative peak is detected. After detecting the full cycle of the walking patterns, the trained CNN model in Python will predict the identity information and send a corresponding command to Unity 3D for the control of door access ("granted/denied") based on the recognized results whether the person is a valid user or not. All the performed experiments in this work complied with a protocol approved by the National University of Singapore Institutional Review Board (N-18-069). All participated subjects were volunteers, and informed consent was obtained prior to participation in the experiments.

## Data availability
The data that support the findings of this study are available from the corresponding author upon reasonable request.

## Code availability
The codes that support the findings of this study are available from the corresponding author upon reasonable request.

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

## Acknowledgements

This work received the financial support from the following grants: the Collaborative Research Project under the SIMTech-NUS Joint Laboratory, "SIMTech-NUS Joint Lab on Large-area Flexible Hybrid Electronics"; the National Key Research and Development Program of China (Grant No. 2019YFB2004800, Project No. R-2020-S-002); and the National Research Foundation Singapore under its AI Singapore Programme (Award Number: AISG-GC-2019-002) "Explainable AI as a Service for Community Healthcare."

## Author contributions

Q.S. and C.L. generated the design concept. Q.S. and C.L. designed the floor mat patterns and overall system architecture. Q.S., B.W., X.S., and B.S. fabricated and assembled the floor mat array. Q.S., Z.Z., and Z.S. built the measurement circuitry and machine learning algorithm. Q.S., Z.Z., T.H., Z.S., B.W., and Y.F. performed all the measurements and data analysis. Q.S. and Z.Z. wrote the manuscript. C.L. supervised the project. All authors reviewed and commented on the manuscript.

## Competing interests

The authors declare no competing interests.
