## [Peer Review File · Nature Communications]

Reviewers' Comments:

Reviewer #1:

Remarks to the Author:

Remarks to the Author:

This manuscript reported a smart floor monitoring system through the integration of self-powered triboelectric sensing mechanism and deep learning data analytics. Throughout the paper, the most innovative aspect described by authors is the combination of deep learning. The idea is interesting and kind of important to the field. Before the manuscript can be accepted by Nature Communications, there are some major issues have been addressed.

- 1.The floor was made by a matrix sensors. Do the performance of all sensors are the same? Will the walking direction change the results?
- 2.In the study, the authors differentiate the output signal by designing TENG units with different electrode coverage rates. However, the triboelectric output and charge collection area are not theoretically linear, so what is your basis for designing such an electrode coverage gradient? Does this gradient design produce good signal discrimination?
- 3.In Fig. 1d, the design of the electrodes does not seem to be regular patterns. If so, what is the merit for designing irregular patterns? For example, if someone 's feet stepped on a floor mat with an electrode coverage rate of 20%, can you make sure that the electrode area which is stepped on accounts for ~20% of the person's foot area?
- 4.Authors should check the typographical errors. For example, Line 44: "large-are".

Reviewer #2:

Remarks to the Author:

The material presented in the paper is of some interest.

- 1) At various places in the paper the English language presentation has to be improved; an editing effort is recommended. (the Abstract is too long and should be simplified)
- 2) Some description of the actual costs of the mats and of an implementation should be provided.
- 3) additional information about the "identification process" (beyond the fact that AI is used) should be provided. What is the reliability of the process? What system 'training' is involved? What are the false positives/false negatives?
- 4) A cost/reliability comparison with other methods (or as a minimum an identification of alternative methods) for people-specific tracking should be provided: person-identification-tracking via smart tags/RFID/UWB sensors/bluetooth-or-Wifi beacons, etc.; additionally a discussion of simple 'presence' sensing (e.g, PIR, photocell, etc.) should be provided.

Publish paper after suggested points are incorporated

Reviewer #3:

Remarks to the Author:

This article proposes a deep learning enabled smart mat system that could be used as a scalable floor monitoring system. The entire system is outlined, including the way in which it was built, results,,,, methods, discussion and evaluation. Detailed supplementary data is provided to support the claims made in the paper. The paper can relatively easily be read and understood and is well-structured.

The main novelties presented in this paper are: (1) the smart mat system itself, and (2) its

connection with a deep learning model to enable detection of end users. These two main novelties are sufficiently novel and will, in my impression, be of interest to others in the community and wider field. Given all the supplementary material, the work is also convincing.

Regarding the contents, I have very few comments. My main comments relate to how the proposed system can be of use in practice. This is at the moment not a core part of the current paper, as the authors (rightfully) focus on the technical detail. Yet, I think it would be of value to expand a bit more on the usage side of the system and the challenges that still exist to broadly apply this system in practice. Below comments give a bit more detail in this regard.

- The main reason to build smart mats is that traditional (often camera-based) techniques are said to raise privacy concerns. I understand that it is problematic to video-record everyone approaching a door because of privacy issues. Yet, in your method, you also claim to recognize individuals (Identities) from their gait. Hence, you can monitor perfectly well who is walking where, even though you do not have the video feed. It seems to me that this does not resolve the privacy problem / your original problem. Besides that you do not have the video recordings, you are still allowing to track a person through a building - which still results in privacy concerns. It would be worthwhile if you could expand on this matter and indicate that this is not entirely solved. It might be that the privacy concern is basically a false claim, in which it would be needed to evaluate to what extent your method provides better detection possibilities than camera-based detection techniques.

- In my opinion, the above comment also connects to the particular use cases that you target and which are only very briefly mentioned in the article (e.g. line 19). Using these smart mats for security cases is very different to their use in elderly homes for fall detection. The first case likely requires identity information, the second maybe not. I think you need to select more specifically instead of presenting your solution as a response to all those use cases. In this regard, it would be great if your introduction and discussion section elaborate on the use case scenarios of your proposed smart mats in greater detail and shows its value in more detail.

- I have my doubts regarding the recognition of individuals through their gait, and I wonder about the 'failsafeness' / trustworthiness of this method. If a door is only supposed to open if a person walks in a certain way, what happens then if this particular person is in a distress situation (e.g. a fire), in which case he runs to the door. This person might not be able to calm down sufficiently in order to be able to walk in the exact good way that would allow the door to open. Is it possible to provide more details about this part? For example, can a test be made in which the same person walks and runs to the door and will the door then still open because the user is recognized? Are such statistics available? It seems to me that you will need to have more test data to evaluate this part of your work.

The article has a small number of typo's which the authors likely want to correct in this phase.

Review made by Pieter Pauwels

Point-by-Point Response to the Reviewers' Comments

Dear Reviewers,

Thank you very much for your careful review, valuable comments, and helpful suggestions regarding our manuscript. We have included new supporting experimental data and revised the entire manuscript carefully according to the comments, and the detailed corrections in a point-by-point manner are listed below.

Reviewer #1:

Comments to the Authors:

This manuscript reported a smart floor monitoring system through the integration of self-powered triboelectric sensing mechanism and deep learning data analytics. Throughout the paper, the most innovative aspect described by authors is the combination of deep learning. The idea is interesting and kind of important to the field. Before the manuscript can be accepted by Nature Communications, there are some major issues have been addressed.

Reply: We thank Reviewer for the careful review and the valuable comments. We have revised the manuscript carefully according to Reviewer's point-by-point comments. Please be noted that the revised portions are marked in red in the revised manuscript.

1. The floor was made by a matrix sensors. Do the performance of all sensors are the same? Will the walking direction change the results?

Reply: Thank you for your comments. Two sets of floor mats with different electrode coverage rates (i.e., 0%, 20%, 40%, 60%, 80% and 100%) are connected into two output electrodes by an interval parallel connection scheme, where the six mats in each set are directly connected together in parallel. The same level of output performance can be achieved for floor mats with the same electrode coverage rate, for example the 0% floor mat from the first set and the 0% floor mat from the second set have the same output performance. But since they are connected to different output electrodes, so they can be differentiated in the output signal patterns. As for the floor mats with different electrode coverage rates, the output gradually increases with increasing the electrode coverage rate, as shown in the characterizations in Fig. 2a-f

(the actual output voltages for floor mats with different electrode coverage rates) and Fig. 2g-i (the increment trends with electrode coverage rate).

In terms of the effect of walking directions, the output voltages from four walking directions with respect the floor mat (N, north; E, east; S, south; W, west) by both right foot stepping and left foot stepping (in total eight cases) have been measured and shown in Fig. 2a-f for the six floor mats, respectively. It can be seen that similar performance can be achieved for all the four walking directions under both right foot stepping and left foot stepping. Thus, for a person walking on the floor mats, the walking direction will not change the results.

Fig. 2. Characteristics of the output performance of individual DLES-mat on a $1\text{ M}\Omega$ external load. **a-f** The generated voltages by repeated stepping motions with PTFE shoe (four directions toward north, east, south and west for both the right foot and the left foot) on DLES-mats with electrode coverage rate of **a** 0%, **b** 20%, **c** 40%, **d** 60%, **e** 80%, and **f** 100%. **g** The corresponding maximum voltage, minimum voltage and peak-to-peak voltage from Fig. 2a-f with respect to different electrode coverage rates, all showing clear increment trends with the increased electrode coverage rate. **h** The effect of different users wearing PTFE shoes on the output peak-to-peak voltage. Similar increment trends of relative magnitudes can be observed from different users, indicating the suitability of the DLES-mat design for individual position sensing. **i**

The effect of different contact materials worn by the same user on the output peak-to-peak voltage, where similar increment trends can also be observed.

2. In the study, the authors differentiate the output signal by designing TENG units with different electrode coverage rates. However, the triboelectric output and charge collection area are not theoretically linear, so what is your basis for designing such an electrode coverage gradient? Does this gradient design produce good signal discrimination?

Reply: Thank you for your comments. According to the triboelectric theory, under the same contact conditions (e.g., same contact area, pressure, etc.), the same amount of charges by triboelectrification should be generated on a dielectric friction surface, and then a larger electrode area beneath the dielectric surface can collect more charges through the electrostatic induction. Since triboelectric device can be normally analyzed by a variable capacitor model, its generated open-circuit voltage can thus be given by $V_{oc} = Q/C$, where Q is the effective induced charges on the electrode (positively related to the electrode area) and C is the equivalent triboelectric capacitance. For our parallel-connection design of multiple triboelectric sensors to minimize the output electrodes, they share the same equivalent capacitance in output generation. In this regard, the triboelectric sensors with different electrode areas will generate electrical outputs with different signal magnitudes, proportional to the effective induced charges on the electrode, which makes them distinguishable in a parallel connection. Therefore, through the design of different electrode coverage rates, output variation in each floor mat and differentiation can then be realized.

A design of 20% difference between two adjacent electrode coverage rates is adopted to achieve rational balance between the distinguishable distinction and number of floor mats from 0% to 100%. As for the layout of electrodes, grid pattern with filled squares is adopted to design the various electrode coverage rates across the relatively large floor mat area. Uniform grid lines throughout the entire floor mat area are designed as the electrode interconnection, forming a 20×20 array of empty squares (each square with a size of $20 \text{ mm} \times 20 \text{ mm}$). Then certain number of squares are selected to be filled with metal to achieve uniformly-distributed electrode coverage rates across the floor mat area (i.e., 0 filled square for 0%, 80 filled squares for 20%, 160 filled squares for 40%, 240 filled squares for 60%, 320 filled squares for 80%, and 400 filled squares for 100%). Although due to the effect of the fringe field, the grid interconnection and the selected squares not perfectly uniform across the whole floor mat area, the designed electrode pattern may not achieve theoretically linear induced charges under each foot stepping. But according to the characterization

results shown in Fig. 2, the signal discrimination among different floor mats with this grid gradient design is satisfactory enough for current sensing applications, where a clear increment trend with sufficient differentiation in the relative signal magnitudes can be observed. In the future, designs of finer grid line spacing and other electrode patterns (such as comb electrode pattern) can be applied to achieve a more uniformly distributed and linear electrode coverage rate throughout the entire floor mat.

3. In Fig. 1d, the design of the electrodes does not seem to be regular patterns. If so, what is the merit for designing irregular patterns? For example, if someone's feet stepped on a floor mat with an electrode coverage rate of 20%, can you make sure that the electrode area which is stepped on accounts for ~20% of the person's foot area?

Reply: Thank you for your comments. The design of the irregular patterns of the 20%, 40%, 60%, and 80% floor mats is mainly based on two considerations. On the one hand, most of the steps will be on the areas closed to the center of the floor mat, and even not, a large portion of the steps will be. Thus, the electrode coverage rates for the middle area (~80%) of the floor mats are designed to be as uniformly distributed and close to the designated values as possible. On the other hand, another merit for designing the current patterns in the 20%, 40%, 60%, and 80% floor mats is that only one printing mask is required for the four floor mats, since the same mask of the 20% floor mat can be used to produce other electrode patterns. First, with the mask oriented upward, the printing will result in the 20% floor mat. Then after obtaining the 20% floor mat, further printing on it with the mask oriented downward will result in the 40% floor mat. Similarly, further printing with the mask oriented right and left will result in the 60% and the 80% floor mats, respectively. The gradual printing results to achieve these four floor mats with different orientations are also shown in Fig. S1 below. Based on these two considerations, the current patterns for the 20%, 40%, 60%, and 80% floor mats are designed. Although for a certain step on the floor mat (for example 20%), the actual electrode coverage rate under the stepping area may not be exactly 20% due to the non-ideal uniformity across the entire floor mat, irregular step shape and random stepping area, but the resultant uncertainty is within acceptable range (indicated by the error bars in the increment trends in Fig. 2) and the measurement results are good enough for the floor monitoring application. In the future, designs of finer grid line spacing and other electrode patterns (such as comb electrode pattern) can be applied to achieve a higher uniformity throughout the entire floor mat to increase the accuracy.

Fig. S1. The gradual printing results using the same printing mask oriented in different directions to achieve the 20%, 40%, 60%, and 80% floor mats. The different filled colours indicate the printing results from each printing.

4. Authors should check the typographical errors. For example, Line 44: “large-are”.

Reply: Thank you for your comments. We have corrected this typo by changing “large-are” into “large-area” in the revised manuscript Page 3. In addition, we have also gone through the whole manuscript carefully to correct other typos and errors.

Reviewer #2:

Comments to the Authors:

The material presented in the paper is of some interest.

Reply: We thank Reviewer for the careful review and the valuable comments. We have revised the manuscript carefully according to Reviewer’s point-by-point comments. Please be noted that the revised portions are marked in red in the revised manuscript.

1. At various places in the paper the English language presentation has to be improved; an editing effort is recommended. (the Abstract is too long and should be simplified).

Reply: Thank you for your comments. We have gone through the entire manuscript carefully to improve the English language presentation and correct the possible errors. Meanwhile, the abstract has also been simplified in a more concise manner.

2. Some description of the actual costs of the mats and of an implementation should be provided.

Reply: Thank you for your comments. According to our estimation on the consumed materials (including Ag, PET, PVC, etc.) and the fabrication expenditure which can be significantly reduced in mass production, the actual cost for one set of the DLES-mats (including 0%, 20%, 40%, 60%, 80%, and 100% mats) is about US\$28.7. Hence the average cost for one DLES-mat with a large sensing area of 40 cm × 40 cm is around US\$4.8. We have included such description of the cost of the floor mats in the revised manuscript's Supplementary Note S1, as well as the implementation approach in the "Methods" section at Page 24.

Page 24: "**Fabrication and implementation of the triboelectric DLES-mats.** A thin layer of polyethylene terephthalate (PET) with relatively high triboelectric positivity is utilized as the friction surface for common foot stepping. The PET is a semi-crystalline polymer film with a high optical transparency, a thin thickness of 125 μm, and a glass transition temperature (T_g) of 81.5 °C. First, to achieve individual floor mat, the large-area PET thin film is cut into a square shape with a dimension of 42 cm × 42 cm. The PET thin film is then pre-treated on one side with a primer treatment for promoting the adhesion with the later printed electrode layer. After that, a layer of silver paste as the charge collection electrode is printed with controlled thickness of about 15 μm on the pre-treated PET surface by screen printing through a designated mask. Next, the screen-printed silver on the PET is cold-laminated with a layer of 80-μm thick polyvinyl chloride (PVC). The PVC layer serves as the supporting substrate with a square opening of 2 cm × 2 cm on the connector pad for wiring purpose. Following that, a copper wire is connected to the electrode by conductive paste through the opening, which is then sealed with thin Kapton tape. Last, different fabricated floor mats are pasted on a woolen floor, and the wires from each floor mat are connected based on the investigated connection scheme for later characterizations."

3. Additional information about the "identification process" (beyond the fact that AI is used) should be provided. What is the reliability of the process? What system 'training' is involved? What are the false positives/false negatives?

Reply: Thank you for your comments. The "training" means the input data updates the weight value of the neural network after the forward propagation and back propagation. The specific setting of the "training" process can be found in the

Methods section “Data collection and deep learning training model” in Page 25, which is also included here.

Page 25: **“Data collection and deep learning training model.** The generated triboelectric signals from the DLES-mat array are acquired by the signal acquisition module in an Arduino MEGA 2560 microcontroller in a real-time manner. In terms of the training data for individual recognition, the signal data from each channel is recorded with 1600 data points (2 channels in total) and 100 samples are collected for each user’s walking pattern, where 80 samples are used for training (80%) and 20 samples are used for testing (20%). A whole dataset is built from 10 different users, with a total number of 1000 samples. The CNN models used in the system are configured as follows: the categorical cross-entropy function is applied as the loss function, Adaptive Moment Estimation (Adam) is used as the update rule due to its optimization convergence rate, and prediction accuracy is used to evaluate the model training. The CNN models are developed in Python with Keras and a tensorflow backend. The feature-based models are trained on a standard consumer-grade computer. Learning rate can be adjusted during training using a Keras callback.”

In addition to the original gait patterns from 10 different users, we have also collected more gait patterns from another 10 new users for testing the result of the class of invalid users (i.e., unrecorded strangers). Thus, the whole test dataset is built from 20 different users in total (2000 samples), and for each user, 20 samples are used for result testing (400 samples). The final prediction results are shown in following table.

TP = 77	FP = 11	88 (predicted positive)
FN = 3	TN = 309	312 (predicted negative)
80 (positive / valid user)	320 (negative / invalid user)	

where “true positive (TP)” is the number of correctly predicted gait patterns for a valid user, “false positive (FP)” is the number of incorrectly predicted gait patterns for an invalid user, “false negative (FN)” is the number of incorrectly predicted gait patterns for a valid user, and “true negative (TN)” is the number of correctly predicted gait patterns for an invalid user.

Then the true positive rate (TPR) and false positive rate (FPR) are calculated as follows:

$$\text{TPR} = \text{TP}/(\text{TP}+\text{FN}) = 96.25\%$$

$$\text{FPR} = \text{FP}/(\text{FP}+\text{TN}) = 3.44\%$$

It can be seen that the prediction result is closed to the upper left corner or coordinate (0,1) of the receiver operating characteristic (ROC) space, which means our method is effective for distinguishing the valid user and invalid user.

4. A cost/reliability comparison with other methods (or as a minimum an identification of alternative methods) for people-specific tracking should be provided: person-identification-tracking via smart tags/RFID/UWB sensors/bluetooth-or-Wifi beacons, etc.; additionally a discussion of simple 'presence' sensing (e.g. PIR, photocell, etc.) should be provided.

Reply: Thank you for your comments. For the people-specific tracking, one of the most popular approaches is based on the use of smart tags. In this monitoring framework, each user needs to wear a tag that advertises its unique identity (ID) based on various wireless communication technologies, such as Wi-Fi¹, Bluetooth², and RFID³. With the smart tag, human identification becomes trivial with a priori knowledge of the linking between the tag ID and the individual. However, these methods based on smart tags always face the problem that the users may be reluctant to wear the sensor or tag due to the suitability⁴. Because of the portability, the tag may be stolen or misused by others, raising extra security concern. In addition, the cost of smart tag based system is usually high, and it normally does not have the position tracking functionality like our smart floor monitoring system. Hence, the non-wearable identification approach is a potential way to address these issues. A comparison table of our method with other non-wearable methods has been provided, as shown in the Table R1 below.

In the SensFloor method⁵, user tracking and localization is obtained based on an array of commercial capacitive sensors placed under the floor. For the WiFi tracking method⁶, user identification from a small group of people can be achieved using the channel state information (CSI) in WiFi to recognize the user's walking steps and walking gait. Then for the floor vibration sensor method⁷, an indoor identification system is developed by extracting the user's gait pattern from the footstep induced structural vibration. In the UWB method⁸, an identification system for residents in a home environment is built with ambient ultrawide band (UWB) sensors installed at entrance area, by a Region of Interest (ROI) extraction approach to monitor the body figure and walking gait of each individual. For the PIR method⁹, a distributed and wireless pyroelectric infrared (PIR) sensor network is constructed for identifying and tracking human targets. Together with the developed empirical mode decomposition

and Hilbert-Huang transform, features of human targets can be extracted in both the time domain and the frequency domain for identification.

Comparing with these methods, our smart floor monitoring system is able to achieve the person-identification-tracking functionality with the low-cost and highly scalable triboelectric floor mat sensors. Even though with the biggest dataset in the number of individuals, a high identification rate of 96% can be achieved. Besides, the triboelectric floor mat sensors can produce self-generated electrical signals under footsteps, which can eliminate the power requirement for sensors. The self-generated electrical signal can also be used as walk-up signal to trigger the operation of the whole system that can be in a “sleep” mode when no one is walking to reduce power consumption. Overall, our smart floor monitoring system enables the personal identification and position sensing without the cameral-based privacy concern and the need of carrying any tags or devices. This not only means that our smart floor monitoring system is a highly convenient detection method, but also means that the personal identity (i.e., dynamic walking gait) cannot be misused by others, unlike the tags/cards/fingerprints can be stolen and/or borrowed. Therefore, our smart floor monitoring system provides a low-cost, scalable, privacy-protected, highly convenient, and highly secure individual-monitoring method toward the smart building/home.

Table R1. Comparison of recent studies on non-wearable human identification system.

Person-identification-tracking methods	Cost	Number of Individuals	Identification Rate
SensFloor⁵ (capacitive sensors)	High	3	84.13%
WiFi⁶	Low	6	80.00%
Floor vibration sensor⁷	High	5	96.50%
UWB⁸	Low	8	88.15%
PIR⁹	Low	3	~90.00%
Our Method	Low (US\$4.8 for a 40 cm × 40 cm mat)	10	96.00%

*UWB: ultra-wide band; PIR: pyroelectric infrared.

Publish paper after suggested points are incorporated

Reply: We thank Reviewer for the careful review and the valuable comments. We have carefully addressed Reviewer's suggestions in a point-by-point manner as shown above, and revised the entire manuscript accordingly.

Reviewer #3:

Comments to the Authors:

This article proposes a deep learning enabled smart mat system that could be used as a scalable floor monitoring system. The entire system is outlined, including the way in which it was built, results,,, methods, discussion and evaluation. Detailed supplementary data is provided to support the claims made in the paper. The paper can relatively easily be read and understood and is well-structured.

The main novelties presented in this paper are: (1) the smart mat system itself, and (2) its connection with a deep learning model to enable detection of end users. These two main novelties are sufficiently novel and will, in my impression, be of interest to others in the community and wider field. Given all the supplementary material, the work is also convincing.

Regarding the contents, I have very few comments. My main comments relate to how the proposed system can be of use in practice. This is at the moment not a core part of the current paper, as the authors (rightfully) focus on the technical detail. Yet, I think it would be of value to expand a bit more on the usage side of the system and the challenges that still exist to broadly apply this system in practice. Below comments give a bit more detail in this regard.

Reply: We thank Reviewer for the careful review and the valuable comments. We have revised the manuscript carefully according to Reviewer's point-by-point comments. Please be noted that the revised portions are marked in red in the revised manuscript.

- The main reason to build smart mats is that traditional (often camera-based) techniques are said to raise privacy concerns. I understand that it is problematic to video-record everyone approaching a door because of privacy issues. Yet, in your method, you also claim to recognize individuals (Identities) from their gait. Hence, you can monitor perfectly well who is walking where, even though you do not have the video feed. It seems to me that this does not resolve the privacy problem / your original problem. Besides that you do not have the

video recordings, you are still allowing to track a person through a building - which still results in privacy concerns. It would be worthwhile if you could expand on this matter and indicate that this is not entirely solved. It might be that the privacy concern is basically a false claim, in which it would be needed to evaluate to what extent your method provides better detection possibilities than camera-based detection techniques.

Reply: Thank you for your comments. We agree with Reviewer that the mentioned privacy concerns should be addressed more appropriately in our manuscript. According to our understanding of the privacy, there are different levels of concerns, e.g., no identity revealed, only identity revealed and no video-recording, identity revealed by video-recording, and biometric privacy data, etc. **First**, most of the people consider that identity revealed by video-recording induces higher concerns than that without video-recording, and normally do not want to be video-recorded. Since for some applications, revealing personal identity is unavoidable such in the security access and other control areas. Thus, to certain extent depending on the applications, our smart floor monitoring system may still need to recognize the personal identity, but it can certainly eliminate the concerns of video-recording. **Second**, compared to the recognition technologies by using smart tags with users, the smart floor monitoring system offers a more convenient way to achieve recognition, since no extra tags or devices require to be carried by the users. **Third**, the smart floor monitoring system is a highly secure technology for personal identity recognition. Since for the recognition technologies such as smart tags and even biometric data-based recognition, the identity information (e.g., tags, cards, devices, fingerprints, etc.) could be stolen and misused by others. On the other hand, the identity information used by the smart floor monitoring system is one's dynamic walking gait, which can hardly be copied by others due to its uniqueness.

For some auto-access applications where no specific identity of the person is required, a possible recognition approach can be implanted to improve the privacy protection. At the training stage for the deep learning model, we do not include the labels with privacy information (e.g., name of the person) for the walking gait-induced output patterns, but only include a label of "valid user" for all the persons with valid access. Thus in the practical scenarios when a person walks through the floor mat array, if his walking pattern is recognized then a message of "valid user" will be displayed without revealing any of his privacy information and the door will be opened. And if his walking pattern is not recognized, then "invalid user" will be displayed and the door will remain closed. In this way, the recognition of valid and invalid users can be achieved without revealing the identity and the privacy information of the person. Accordingly, we have revised all the descriptions talking about the privacy concerns

in the manuscript and have also included additional discussions to improve privacy protection in Page 22.

- In my opinion, the above comment also connects to the particular use cases that you target and which are only very briefly mentioned in the article (e.g. line 19). Using these smart mats for security cases is very different to their use in elderly homes for fall detection. The first case likely requires identity information, the second maybe not. I think you need to select more specifically instead of presenting your solution as a response to all those use cases. In this regard, it would be great if your introduction and discussion section elaborate on the use case scenarios of your proposed smart mats in greater detail and shows its value in more detail.

Reply: Thank you for your comments. In terms of the security access as demonstrated, both the required person identity and walking position can be obtained from the smart floor monitoring system by detecting output voltage peaks and recognizing his walking gait related output patterns. Thus, automatic lighting control and security access can be achieved in the demonstration. For fall detection, only the position sensing information is required, since the output signals will be regularly generated for normal walking. But when somebody falls on the floor mat array, multiple output peaks will be generated in a short period due to falling motion induced rapid contacts and no outputs are generated after that. The typical signal patterns for the normal walking process and the walking-falling process on the floor mat array are shown in Fig. S6, which is also included here. Hence, fall detection can be realized by the floor mat system itself through time-domain signal monitoring. In terms of healthcare application, the floor mat system can be used for exercise monitoring, including the type of exercise (e.g., walking, slow/normal/fast running, jumping, etc.), the time period of the exercise, and the burned calories based on the type and period of the exercise. We have also included more detailed information in the introduction and results section in the revised manuscript.

Fig. S6. The typical signal patterns for **a** the normal walking and **b** the walking-falling process on the floor mat array.

- I have my doubts regarding the recognition of individuals through their gait, and I wonder about the 'failsafeness' / trustworthiness of this method. If a door is only supposed to open if a person walks in a certain way, what happens then if this particular person is in a distress situation (e.g. a fire), in which case he runs to the door. This person might not be able to calm down sufficiently in order to be able to walk in the exact good way that would allow the door to open. Is it possible to provide more details about this part? For example, can a test be made in which the same person walks and runs to the door and will the door then still open because the user is recognized? Are such statistics available? It seems to me that you will need to have more test data to evaluate this part of your work.

Reply: Thank you for your comments. In order to demonstrate the applicability of the smart monitoring system in various situations, additional testing with the same person (4 different users) in different statuses (i.e., normal walking, fast walking and running) has been conducted. The corresponding signal patterns and the predicted results are shown in Fig. S11 and Fig. S12, respectively. The signal data of each class is recorded with 1600 data points (2 channels in total) and 100 samples are collected where 60 samples are used for training (60%), 20 samples are used for validating (20%) and 20 samples are used for testing (20%).

Using the same CNN structure to train the dataset, the newly trained DL model can distinguish the different passing statuses of 4 users (total 12 classes) with an accuracy of 89.17%. In addition, if all the three passing statuses from the same user is set as one individual label (just distinguish the user without knowing his walking status), the accuracy of the testing set after training reaches 91.47%. These results show that even when the user passes through the DLES-mat array in different ways, the smart floor monitoring system can still recognize and identify the user with a high accuracy of 91.47%.

Fig. S11. The generated voltage patterns through the middle row of the DLES-mat array with different passing statuses (i.e., normal walking, fast walking, and running). **a-c** The output voltage patterns from User 1 (U1). **d-f** The output voltage patterns from User 2 (U2). **g-i** The output voltage patterns from User 3 (U3). **j-l** The output voltage patterns from User 4 (U4).

Fig. S12. The confusion matrix for individual recognition in different statuses with **a** known passing status, and **b** unknown passing status.

The article has a small number of typo's which the authors likely want to correct in this phase.

Reply: Thank you for your comments. We have double checked the entire manuscript carefully to correct the potential typos and errors in the revised manuscript, where the corrections are are marked in red.

Reference:

1. Zhang J, Wei B, Hu W, Kanhere SS. WiFi-ID: Human identification using WiFi signal. *Proc - 12th Annu Int Conf Distrib Comput Sens Syst DCOSS 2016*. 2016:75-82.
2. Alhamoud A, Nair AA, Gottron C, Böhnstedt D, Steinmetz R. Presence detection, identification and tracking in smart homes utilizing bluetooth enabled smartphones. *Proc - Conf Local Comput Networks, LCN*. 2014;2014-November(November):784-789.
3. Helfenbein T, Király R, Töröcsik M, Tóth E, Király S. Extension of RFID based indoor localization systems with smart tags. *Infocommunications J*. 2017;9(3):25-31.
4. Srinivasan V, Stankovic J, Whitehouse K. Using Height Sensors for Biometric Identification in Multi-resident Homes. In: Floréen P, Krüger A, Spasojevic M, eds. *Pervasive Computing*. Berlin, Heidelberg: Springer Berlin Heidelberg; 2010:337-354.
5. Sousa M, Techmer A, Steinhage A, Lauterbach C, Lukowicz P. Human tracking and identification using a sensitive floor and wearable accelerometers. *2013 IEEE Int Conf Pervasive Comput Commun PerCom 2013*. 2013;(March):166-171.
6. Zeng Y, Pathak PH, Mohapatra P. WiWho: WiFi-Based Person Identification in Smart Spaces. *2016 15th ACM/IEEE Int Conf Inf Process Sens Networks, IPSN 2016 - Proc*. 2016.
7. Pan S, Wang N, Qian Y, Velibeyoglu I, Noh HY, Zhang P. Indoor person identification through footstep induced structural vibration. *HotMobile 2015 - 16th Int Work Mob Comput Syst Appl*. 2015:81-86.
8. Mokhtari G, Zhang Q, Hargrave C, Ralston JC. Non-Wearable UWB Sensor for Human Identification in Smart Home. *IEEE Sens J*. 2017;17(11):3332-3340.
9. Xiong J, Li F, Zhao N, Jiang N. Tracking and recognition of multiple human targets

moving in a wireless pyroelectric infrared sensor network. *Sensors (Switzerland)*.
2014;14(4):7209-7228.

Reviewers' Comments:

Reviewer #1:

Remarks to the Author:

Authors have addressed all questions that reviewers concerned, I do not have further questions and recommend to accept this manuscript by Nature Communications.

Reviewer #2:

Remarks to the Author:

I have re-looked at the paper.

I see that an effort has been made to incorporate the technical feedback.

However, the paper needs another (quick) round to improve the English, even in the abstract. In particular and as an example the phrase "based on triboelectric mechanism" is used a number of times. It would be better to say "based on a triboelectric mechanism" or "based on the triboelectric mechanism" or "based on triboelectric mechanisms"... several other linguistic improvements are urged. Technical content deserves publication, but language needs some cleanup.

Sincerely,

Reviewer #3:

Remarks to the Author:

My earlier review comments have been appropriately addressed. I am satisfied with the changes made regarding the potential use cases, as well as the clarifications regarding privacy concerns.

Although language has improved, there are still a number of language errors. Please try to let this article be proofread by a native speaker.

Other than the language, this paper is ready for publication in my opinion.

Point-by-Point Response to the Reviewers' Comments

Dear Reviewers,

Thank you very much for your careful review, valuable suggestions, and positive comments regarding our manuscript. Based on Reviewers' suggestions, we have asked a native speaker colleague to check the entire manuscript carefully and then revised it accordingly. The revised portions are **marked with the "track changes" feature** in the revised manuscript, and the detailed responses in a point-by-point manner are listed below.

Reviewer #1 (Remarks to the Author):

Authors have addressed all questions that reviewers concerned, I do not have further questions and recommend to accept this manuscript by Nature Communications.

Reply: We thank Reviewer for the careful review and positive comments on our work.

Reviewer #2 (Remarks to the Author):

I have re-looked at the paper.

I see that an effort has been made to incorporate the technical feedback.

However, the paper needs another (quick) round to improve the English, even in the abstract. In particular and as an example the phrase "based on triboelectric mechanism" is used a number of times. It would be better to say "based on a triboelectric mechanism" or "based on the triboelectric mechanism" or "based on triboelectric mechanisms" ... several other linguistic improvements are urged. Technical content deserves publication, but language needs some cleanup.

Reply: We thank Reviewer for the careful review and helpful suggestions on our work. We have revised the statement of "based on triboelectric mechanism" to "based on the triboelectric mechanism" in the manuscript. Then we have also performed exhaustive linguistic improvements throughout the manuscript by going into the details carefully with a native speaker colleague.

Reviewer #3 (Remarks to the Author):

My earlier review comments have been appropriately addressed. I am satisfied with the changes made regarding the potential use cases, as well as the clarifications regarding privacy concerns.

Although language has improved, there are still a number of language errors. Please try to let this article be proofread by a native speaker.

Other than the language, this paper is ready for publication in my opinion.

Reply: We thank Reviewer for the careful review and positive comments on our work. To improve the language throughout the manuscript, we have asked a native speaker colleague to proofread the entire manuscript carefully and revised it accordingly.